# Problematic Smartphone Use Leads to Behavioral and Cognitive Self-Control Deficits

**DOI:** 10.3390/ijerph19127445

**Published:** 2022-06-17

**Authors:** Rosa Angela Fabio, Alessia Stracuzzi, Riccardo Lo Faro

**Affiliations:** Department of Economy, University of Messina, 98121 Messina, Italy; alessiastracuzzi98@gmail.com (A.S.); riccardolofaro97@gmail.com (R.L.F.)

**Keywords:** smartphone addiction, self-control deficits, cognitive effects, smartphone withdrawal

## Abstract

Excessive use of smartphones has been associated with a number of negative consequences for individuals. Some of these consequences relate to many symptoms of behavioral addiction. The present study aims to investigate whether participants with high levels of smartphone usage may have difficulty with their ability to wield the self-control that is needed to restrict smartphone usage compared to participants with lower levels of smartphone addiction. Specifically, we expect that people with high levels of smartphone usage may have problems in refraining from using a smartphone. In addition, we expect people with a high level of smartphone use may show deficiencies in cognitive tasks such as memory, executive control, and visual and auditory attention. An ABA design was applied to analyze the effects of smartphone withdrawal. The first A refers to baseline measurements: Visual RT, Auditory RT, Go/No-Go RT and N-Back RT and Eriksen flanker RT. The B refers to 3 days of smartphone withdrawal, whereas the second A refers to the same measurements used in the baseline. In addition, several standardized scales were administered, among them: Smartphone addiction scale-short version (SAS-SV), Fear of missing out scale (FoMOs), Procrastination scale, and Psychological General Well-Being Index. One hundred and eleven participants took part in the study. Based on median split they were divided into two groups: high level and low level smartphone users. Moreover, thanks to an app installed on the participants’ smartphones, it was possible to measure levels of compliance with the task. Results indicate that participants with low levels of smartphone usage show less difficulty in their ability to wield the self-control needed to withdraw smartphone use and faster reaction times on cognitive tests than participants with high levels of smartphone usage. Moreover, the profile of participants with high levels of smartphone usage shows higher scores on the FoMOs and Procrastination scale, and lower scores in the Psychological General Well-Being Index. The results are discussed in light of self-regulation theory.

## 1. Introduction

From 2007, since the launch of the iPhone by Steve Jobs, the sale of smartphones has increased exponentially. This is reflected in the growing number of smartphone users. Currently there are very few people in industrialized countries who do not have a mobile phone. Recent statistics suggest that over 6.6 bn people around the world use a smartphone to communicate, surf the web or just play video games [1]. The widespread availability of smartphones has led to their overwhelming use in the world, leading to interest from researchers. In recent years, tests have been conducted in an attempt to investigate various phenomena such as TV addiction [2], excessive use of computer games [3], gambling disorders over the web [4] and, generally, internet addiction [5]. The excessive use of smartphones, also known as ‘nomophobia’, has proven to be a form of technological addiction that is rapidly becoming a major social problem around the world [6,7,8,9,10]. Today, addiction is defined as a pleasure-inducing behavior that through repeated exposure gradually leads to loss of control and further negative consequences [11].

Mahapatra [12] found that loneliness and self-regulation deficits are the main antecedents for smartphone addiction, and family and personal conflicts and poor academic performance are the significant negative consequences of excessive use of smartphones. A systematic review on problematic smartphone use (PSU) [13] suggests that people who are young, female, and highly educated are more prone to PSU. Literature [14,15,16] has also shown the effects of smartphone use in everyday life situations. It has negative effects on sleep when used in the bedroom [17], and has negative effects on procrastination when used in the classroom [18]. Finally, it puts the driver and pedestrians at risk when used while driving [18,19,20].

### 1.1. Smartphone Addiction

Smartphone addiction is similar to most addictive disorders, but the fact that a smartphone is small, easily handled and portable makes the risks more insidious and pervasive [21,22]. As with many other forms of addictive behaviors, learning mechanisms play a central role. Duke & Montag [23,24] outline conditioning principles that likely contribute to PSU. Since these behaviors become automatic and therefore largely unconscious, they are hard to interrupt. However, conditioning principles, such as those described in their paper, work better with some people than with others [23,24]. Those who are vulnerable to intermittent reinforcement principles may be affected at a higher level. In this context, the smartphone can be compared with a slot machine. The slot machine does not reward a person every time he/she pushes a button; instead, the individual is rewarded from time to time, without any discernible pattern [24]. The same process can be observed with the smartphone. Smartphones reward us on an intermittent, unpredictable level, with funny messages via WhatsApp or e-mails, leading to a robust smartphone usage pattern [24].

### 1.2. Smartphone Withdrawal

In people that show excessive smartphone use, smartphone restriction might elicit negative effects for certain individuals. These negative effects may be regarded as withdrawal symptoms traditionally associated with substance-related addictions [11]. To address this timely issue, Eide et al. [11] examined 127 participants aged 18–48 years, assigned to one of two conditions: a restricted condition or a control condition. During the 72 h of smartphone restriction, participants completed the aforementioned Smartphone Withdrawal Scale (SWS), the Fear of Missing Out Scale (FoMOS) and the Positive and Negative Affect Schedule (PANAS) three times a day [11]. Their results showed that participants in the smartphone restriction condition increased withdrawal symptoms and a fear of missing out. In general, the negative consequences of excessive smartphone use refer to symptoms such as withdrawal and hindered user productivity, social relationships, physical health, or emotional well-being in daily life [25,26].

### 1.3. Theories on Smartphone and Internet Addiction

Moretta et al. [27] recently published a detailed review on the problematic use of smartphones and the internet. The authors suggested to theoretically integrate smartphone-related problematic behaviors with internet problematic behaviors as, in this way, behavior can be focused on and not the device itself. In line with this, the first author who proposed a cognitive-behavioral model of pathological or problematic internet use (PIU) was Davis [28]. In his model, there are two components that act: one is distal, namely the individual’s psychopathology, and the other is proximal, i.e., maladaptive cognitions associated with internet usage [28]. Caplan [29] later proposed a revision of Davis’ model and included some cognitive/behavioral variables, such as the preference for online social interactions, related to negative outcomes associated with internet and smartphone use. Specifically, he found that communicating through a device reduces the distress triggered by face-to-face social interactions but led to defective self-regulation [29]. Such poor self-regulation would in turn generate negative consequences in the lives of people. After this model, Brand et al. [30] proposed that impaired functioning of prefrontal control mechanisms would be associated with defective self-regulation and coping strategies, which would lead individuals to turn to the online world. People with low levels of executive functioning are vulnerable to intermittent reinforcement principles connected to internet-related activities. These processes are key factors in the transition from voluntary/goal-directed actions (with an appraisal of action consequences) to habitual actions (seemingly automatic and uncontrolled), which is at the basis of addictive behaviors [31,32,33,34,35,36,37,38]. Another model, the Interaction of Person-Affect-Cognition-Execution (I-PACE), proposes that PIU behaviors may be explained by looking at interactions between predisposing factors (e.g., impulsivity, anxiety, depression, general distress), moderators (e.g., coping style, self-regulatory capacities, and internet-related attentional and cognitive biases), and mediators such as reduced inhibitory control in combination with reduced executive functioning and diminished decision making [39,40,41]. As previously mentioned, due to accessibility and portability, the connection between the internet and smartphone use is more insidious and pervasive [21,22]. In line with Duke & Montag [23], we think that participants with high levels of smartphone addiction may have automatized behaviors that have become automatic and therefore largely unconscious; for this reason, they are hard to discontinue. Given this, we will consider the relationship between PSU and withdrawal symptoms, both at behavioral and cognitive levels [42]. 

### 1.4. The Present Study

In line with theories that postulate that PSU and poor self-regulation behaviours are related, the main aim of the present study is to investigate the relationship between smartphone usage and behavioral and cognitive self-control deficits. This objective stems from the observation of young people who have difficulty in putting aside their smartphones and refraining from using them, even for an hour [43]. 

The present study differs from previous ones for two reasons. The first is that self-report questionnaires have been most commonly used to assess the time spent on smartphones and the characteristics of smartphone addiction, and some of these characteristics are identical to the components of all variants of Internet addiction [43]. The results of factor analysis showed that smartphone addiction consists of four main components: compulsive behaviors, tolerance, withdrawal, and functional impairment [21,22]. However, the use of a self-report questionnaire can be a limitation, as we know people tend to overestimate or underestimate usage time and the number of times of app usage. To overcome this limitation, in the present study we used SocialStatsApp. This app objectively measures the frequency of access to each of four applications (WhatsApp, Instagram, Tik Tok and Facebook) and the daily number of minutes of use of each of these four applications by installing it on the participants’ smartphones. The development of the app was possible thanks to the official Integrated Development Environment (IDE) of Android applications, Android Studio. In this way, both objective and subjective measurements of smartphone usage could be detected. 

The second difference, compared to the work by Eide & Arestad [11], is that in this study the limitation of smartphone use was applied in a different way. The participants were requested to maintain just one-hour smartphone use per day for three days. Unlike previous research, in which the cell phone was taken away and locked up, in this study the subjects were given the possibility to keep their smartphones, asking them to exercise self-control over use. They were told that true compliance with the task was assessed during the three-day smartphone restriction phase through the installation of the SocialStatApp. Objective measurements of the time spent on social media use allowed us to measure the level of compliance of the experimental group. 

The main hypothesis of the present study was to investigate whether higher levels of smartphone use in some participants might lead to a higher level of failure of the self-control ability in stopping smartphone use than in participants with lower levels of smartphone use. In particular, we expected people with high levels of smartphone use to fail to intentionally not use the smartphone and, consequently, to show a lack of task compliance. 

Moreover, with reference to studies on cognitive deficiencies and smartphone use, we expected people with high levels of smartphone use to show deficiencies in the following tasks: working memory n-back test, visual reaction time, auditory reaction time, visual reaction time, Go/No-go test and Eriksen flanker test.

Finally, in relation to results of previous studies [11], participants with high levels of smartphone use may show a lower Psychological General Well-Being Index (PGWBI), a higher FoMO score and Procrastination score (PS) than participants with low levels of smartphone use. 

## 2. Method

### 2.1. Participants

One hundred and eleven participants (65 women and 46 men), mean age 32 years (SD = 12.34; range from 18 to 65), were studied. Twenty-eight percent were university students and 72% were workers. 

The sample was recruited through ads published on various platforms, including WhatsApp, Facebook, and Instagram. To participate in the research, participants had to have an Android device, to be able to download the application developed specifically for the experiment and be between 18 and 65 years old. The ads posted on WhatsApp groups were viewed by 159 people; of these, 57 subjects responded but 12 participants, when they were informed, they had to limit the use of their smartphones for a short period of time, did not give their availability (10 computer science students). On Facebook, the ads received 1682 views, among which 56 people responded and only 42 participated. Finally, the post published on Instagram obtained 3000 views with 27 responses and 24 actual participations in the research.

### 2.2. Link between Hypotheses and Research Objectives 

In relation to the main hypothesis that people with high levels of smartphone use may fail to reduce the time spent on the smartphone, participants were divided by median split into two groups in order to identify those with low levels (below median) and high levels (above median) of smartphone use and calculated their mean of minutes of use in three phases: pre-test phase, experimental phase (with smartphone limitation) and post-test phase. 

In relation to the second hypothesis that people with high levels of smartphone use may show deficiencies in cognitive tests, we submitted the two groups to working memory n-back test, visual reaction time, auditory reaction time, visual reaction time, Go/No-go test and the Eriksen flanker test.

In relation to the third hypothesis that people with high levels of smartphone use may show a lower PGWBI, a higher FoMO score and PS than participants with low levels of smartphone use, we asked both groups to fill in the related questionnaires.

### 2.3. Materials

#### 2.3.1. SocialStatsApp

##### Technology Analysis

The ‘SocialStatsApp’ application was created for Android devices, the Operating System (OS) for mobile devices launched by Google at the end of 2007, which holds over 70% of the Worldwide mobile OS Market Share. Regarding iOS, the OS system for mobile devices manufactured by Apple Inc., it was not possible to develop a similar app since Apple, for privacy reasons, does not allow the collection of usage statistics. The app was developed in Java, a high-level programming language, object-oriented, created in 1991 to program home devices (televisions, telephones, etc.). Related in particular to languages such as C++, the basic syntax has been kept almost identical to the latter but “modern” features have been added, such as concurrency support (multithreading) and automatic memory management (garbage collector). Today, Java is still one of the most used programming languages for Client-Server applications. The mobile application has the following main features. 

User authenticationUsage acquisitionStatistics submission

It was designed around the concept of ease of use and privacy. Indeed, only the data of TikTok, Facebook, Instagram and WhatsApp were collected and saved. The app develops in three main stages: (1)Login: Each participant was given a user id, upon startup, at the login stage;(2)Access to statistics: the SocialStats app needs a special permission to function properly (android.permission.PACKAGE_USAGE_STATS) which must be manually granted by the user following the scheme below (Figure 1):

Sending Statistics: the user must send usage statistics to the Backend system based on the timing, 3 or 7 days, as requested by the examiner.

Usage Statistics where collected through the android.app.usage.UsageStatsManager Class added in API level 21. Given this technical limit, it was possible to include only participants with an android phone, with at least Android Lollipop (5.0). 

The API provides access to device usage history and statistics. Usage data is aggregated into time intervals and grouped by package name:TikTok: com.zhiliaoapp.musicallyFacebook: com.facebook.katanaInstragram: com.instagram.androidWhatsapp: com.whatsapp

For Login and data submission, the software interacts with Firebase, the serverless platform supported by Google, which provides a suite of tools for writing, analyzing and maintaining cross-platform applications. In particular, Firestore, an NoSQL database, was used for archiving the extracted data. 

The final step, once the data was submitted by all users, was to extract the data from the database. This step was carried out through a script in Python which allowed aggregation by the user and was later saved in a more readable format for the examiner. The parameter concerned the number of minutes of daily use of each of the four applications (Figure 2).

#### 2.3.2. General Questionnaire 

Some demographic and psychological variables of the participants were first collected: age, gender, measures of subjective well-being, positive and negative emotions, friendship satisfaction, social support, communication measures and networks (frequency of communication with friends on a typical day), impulsivity, anxiety (social anxiety) and depression, awareness related to PSU, level of self-esteem, academic, university and work performance.

##### Smartphone Addiction Scale-Short Version (SAS-SV)

SAS-SV is useful to identify subjects with a high risk of addiction and to estimate the level of severity [44]. The questionnaire includes 10 questions related to disorder of daily life, positive anticipation, abstinence, cyberspace-oriented relationships, excessive use of smartphones and tolerance. For each question, participants express their opinion on a 6-point scale ranging from 1 (strongly disagree) to 6 (strongly agree). The cut-off score level differs by gender. Men are identified at high risk of addiction with scores between 22 and 31, whilst women with scores between 22 and 33 are defined at high risk of addiction [44]. SAS-SV also demonstrated good internal consistency with an alpha reliability of 0.79.

##### Psychological General Well-Being Index

The short version of the PGWBI was used in this study [45]. It is a measure of the level of subjective psychological well-being. Specifically, it evaluates representations of self and intrapersonal affective or emotional states that reflect a subjective perception of well-being. The PGWBI consists of 22 standardized items, with 6 items for the short form. The tool produces a unique measure of psychological well-being. The global measure evaluates the following items: anxiety, depression, positive well-being, self-control, overall health, and vitality. The questionnaire was validated in Italy by Grossi et al. [45].

##### Fear of Missing out Scale

The Fear of Missing Out Scale (FoMOs) [46] refers to the fear that others are having a rewarding experience without us, it is a feeling of being “left out”; it literally means “fear of being cut off” and can be considered a new form of social anxiety. FoMOs consists of 10 items, for example “I’m afraid that others have more rewarding experiences than mine”. The questions are evaluated on a five-point Likert scale, where 1 corresponds to the answer “not at all true for me” and 5 to the answer “extremely true for me”. FoMOs demonstrated good internal consistency with an internal consistency of alpha values ranging from 0.87 to 0.90. 

##### Procrastination Scale

To measure procrastination, the Procrastination Scale (PS) designed in 1986 by Clarry Lay [47] was used. In its original version, it consists of 15 items with a 7-point Likert scale response (from Absolutely disagree to Absolutely agree). PS demonstrated good internal consistency with an internal consistency of alpha values ranging from 0.78 to 0.89.

### 2.4. Procedure

The study design had three different phases: a pre-test, an experimental and a post-test phase (Figure 3).

In the pretest assessment, subjects were asked to download and install an application, specifically developed for this study, on their smartphone. This allowed us to measure a baseline time of use during the seven days before the start of withdrawal. 

Subsequently, during the experimental phase, the participants were asked to limit the use of their smartphone for 72 h (excluding work and/or university reasons). Specifically, the task required participants to reduce the use of their mobile phone to 1 h a day, leaving the possibility for the subjects themselves to best distribute the time of use of the device during the day. They were again aware of the measurements of the installed app.

Finally, in the post-test assessment phase, the time of use during the seven days after the end of the smartphone withdrawal phase was measured. 

In the day before the start of the experimental phase and the day after the end of the experimental phase, cognitive functions were also evaluated through tests carried out on the “Cognitive fun” platform (http://urlm.it/www.cognitivefun.net, accessed on 1 February 2021), including: working memory, attention and executive control. The following tests were used to measure cognitive performance: auditory reaction time (RT auditory attention), visual reaction time (RT visual attention), Go/No go (ability to inhibit motor response), Eriksen flanker test (behavioral inhibition) and n-back (working memory). 

### 2.5. Statistical Analysis

Data were analyzed using SPSS Version 24.0 for Mac. Measurement parameters varied according to the type of task; with reference to the use of a smartphone, the time spent on the smartphone, the frequency of access to each of the four applications (WhatsApp, Instagram, TikTok and Facebook) and the daily number of minutes of use of each of the four applications, were all recorded by the SocialStatsApp, which was installed on the participants’ smartphones. Reaction Times (RT) were used for cognitive task measurement. Parameters were first analyzed with reference to the frequency of compliance to the task and then underwent a 2 × 3 mixed analysis of variances, with Group (smartphone addiction: high levels vs. low levels) as between-subject factor and phases (pre-test, experimental phase, and post-test) as within-subjects factors. Descriptive statistics of the dependent variables were tabulated and examined. RT data referring to correct replies were cleaned, and outliers were removed (outliers referred to more than 1000 ms or less than 200 ms and accounted for 2% of responses). The alpha level was set to 0.05 for all statistical tests. In the case of significant effects, the effect size of the test was reported. For ANOVA, partial eta-squared was used. The Greenhouse–Geisser adjustment for non-sphericity was applied to probability values for repeated measures.

## 3. Results

Data were analyzed with reference to the above-mentioned hypothesis. Before applying the repeated measures analysis of variance, the median of SAS-SV was computed. Participants were divided by median split into two groups to identify those with low levels (below median) and high levels (above median) of smartphone addiction (Iacobucci et al., 2015). 

### Smartphone Addiction and Self-Control

With reference to the first research hypothesis, that was to investigate whether participants with higher levels of smartphone addiction might fail more in the self-control ability of resisting using their smartphones than participants with lower levels of smartphone use, participants were divided into two groups: participants who showed compliance with higher self-control (those with a mean score of less than 1 h) and participants who did not show compliance; thus, with lower self-control (those with a mean score greater than 1 h). Contingency table analysis was then applied to explore relationships between the two groups with low and high levels of smartphone addiction and the two groups with or without compliance. Chi square was significant (χ^2^ = 4.93, df = 72, *p* < 0.02). As can be seen from Figure 4, participants with high levels of smartphone addiction showed a higher percentage of noncompliance.

This hypothesis was also verified with the parameter of minutes of time spent on the smartphone. Table 1 shows the means and standard deviation of minutes of time spent on the smartphone. 

A repeated measures analysis of variance was applied: 2 (group: low level of smartphone addiction vs. high level of smartphone addiction) × 3 (phases: pre-test, experimental phase, post-test). 

Group factor shows a significant effect, F (1, 111) = 294.62, *p* < 0.001, η2p = 0.890; this indicates that participants with high levels of smartphone addiction spent more time than participants with low levels of smartphone addiction. Phase factor shows a significant effect, F (2, 212) = 55.65, *p* < 0.001, η2p = 0.890; this result means that all participants behave differently in relation to the three phases: they spent more time in the pre- and post-test phases, but spent less time during the experimental phase, when participants were requested to exercise self-control ability in suspending smartphone use (Figure 5). 

The second hypothesis was that people with high levels of smartphone addiction may show deficiencies in the following tasks: working memory n-back test, visual reaction time, auditory reaction time, visual reaction time Go/No go test and the Eriksen flanker test.

Table 2 show the means and standard deviations of each cognitive parameter before and after the experimental phase. 

Regarding Go/No go RT, the group factor shows significant effect, F (1, 111) = 76.62, *p* < 0.01, η2p = 0.59; this result indicates that participants with low levels of smartphone usage show faster RTs than participants with high levels of smartphone usage; with reference to the Eriksen Flanker test RT, the group factor again shows significant effect, F (1, 111) = 83.52, *p* < 0.01, η2p = 0.62; this result indicates that participants with low levels of smartphone usage again show faster RTs than participants with high levels of smartphone usage. In all other tests, there were no significant differences.

With reference to the third hypothesis, Table 3 shows the means (and standard deviations) of PGWBI, FoMOs and PS of the 2 groups: low levels of smartphone usage and high levels of smartphone usage. The results confirm that the participants with high levels of smartphone usage show lower PGWBI (t (111) = 2 0.48, *p* < *0*.004), and higher FoMOs (t (111) = 4.34, *p* < *0*.001) and PS (t (111) = 3.48, *p* < *0*.001) than participants with low levels of smartphone usage. 

## 4. Discussion

The main objective of the present study was to investigate whether participants with high levels of smartphone usage might have difficulty in self-control compared to participants with low levels of smartphone usage. The results show that participants with high levels of smartphone use do not show high levels of compliance to the task. As a result, we can say that subjects with high levels of smartphone use show less self-control in smartphone use and spend more time on the phone than participants with low levels of smartphone use. This lack of self-control is intrinsically linked to PSU. Indeed, a consistent body of research has shown that low levels of self-control not only predict high-frequency usage of smartphones, but as evidenced by Wilmer & Chein [48] and Berger et al. [49], there may be a link to smartphone addiction, such as withdrawal symptoms, mood changes and cyberspace-oriented relationships [28,31,32]. However, both groups spent more time on the phone in the pre- and post-test phases than during the experimental phase, when participants were required to wield self-control in suspending smartphone use. In line with Davis et al. and Caplan et al. [28,29], we found that communicating through a device led to defective self-regulation. Such poor self-regulation could in turn generate negative consequences in the lives of people [29].

The first consequence, in line with the second hypothesis, namely that people with high levels of smartphone usage may show deficiencies in cognitive tasks, indicates that in higher-level cognitive tests participants with low levels of smartphone usage have reaction times that are faster than participants with high levels of smartphone addiction. Specifically, the tests showed significant differences between the groups in the Eriksen flanker test and in the visual reaction time Go/No go, whereas in all other cognitive tests no significant differences emerged. These results are in line with the study by Lim [39] who found that smartphone addiction, executive function deficiencies, and memory impairment were positively correlated. 

Finally, from the administration of standardized scales, different profiles of people belonging to the two groups emerged. People with low levels of smartphone use seem to display fewer procrastination behaviors and less fear of being excluded from the online flow of information. In addition, these people have a better perception of their general well-being and quality of life, whereas individuals with high levels of smartphone addiction not only have a worse perception of their own well-being and quality of life, but also show greater procrastination behaviors and fear of being excluded from the flow of information online. The Interaction of Person-Affect-Cognition-Execution can explain the present results, as problematic internet behaviors are highly related to moderators such as self-regulatory capacities, and internet-related attentional and cognitive biases [32].

With respect to the results of Duke & Montag [23] and Eide et al. [11] only in appearance do our results diverge from theirs. In the study by Eide et al. [11], there were precise restrictions imposed on smartphone use and this increased withdrawal symptoms and FoMO. Our results differ as the participants with high levels of smartphone use showed high levels of self-control deficit but, in any case, they could access their smartphones. If they had shared the same restriction as the first study, it may be that the same results could have been achieved. It is interesting to underline that the self-control deficit may predict withdrawal symptoms. Future research may use this as a predictor.

In line with Duke & Montag [23], we think that participants with high levels of smartphone use may have automatized behaviors that have become automatic and therefore largely unconscious; for this reason, they are hard to interrupt. However, conditioning principles, such as those described, will work better with some people than with others [23,50]. People who are vulnerable to intermittent reinforcement principles may be affected to a greater extent. In our study, participants with high levels of smartphone use not only had a worse perception of their own well-being, but also showed greater procrastination behaviors and fear of being excluded from the flow of information online. These factors may determine vulnerability to intermittent reinforcement principles at a higher level as predicted by the I-PACE model [32]. Smartphones, indeed reward us on an intermittent, unpredictable level with funny messages via WhatsApp or e-mails, leading to a robust smartphone usage pattern.

## 5. Limitation

A limitation of the present study relates to the sample. As seen in the participant section, subjects that saw the ads posted on WhatsApp groups, on Facebook sites and on Instagram were much higher in number than the actual participants in the research. Some subjects, on knowing that there would be a request to withdraw their smartphone and to resist the use of the device, immediately refused to adhere to the study; therefore, it is feasible to think that we do not have the data of subjects who are most strongly addicted to smartphone use, and this could mean that the effects may be stronger than we found. Indeed, participants with high levels of smartphone use were not able to refrain from usage, and this may be useful in highlighting withdrawal symptoms.

Therefore, future research should examine a larger sample of participants including more highly addicted participants. The complete range of attentional capacities might also be analyzed through diverse attention tasks: general executive and distributed attention tasks. 

## 6. Conclusions

In conclusion, the results that emerged from the present research are in line with the hypotheses. Specifically, as expected, participants with high levels of smartphone use showed less self-control and deficits in cognitive tasks (Eriksen Flanker test, Go/No-go visual reaction time) compared to those with low levels of smartphone use. It was possible to profile participants with high levels of smartphone use compared to those with low levels of use. People with high levels of addiction show procrastination behaviors and fear of being excluded from the flow of information online and, moreover, have worse perceptions of their own well-being and quality of life than participants with low levels of addiction. Therefore, from this study, it emerges that people with high levels of smartphone use show difficulties in behavioral and cognitive self-control.

## Figures and Tables

**Figure 1 ijerph-19-07445-f001:**
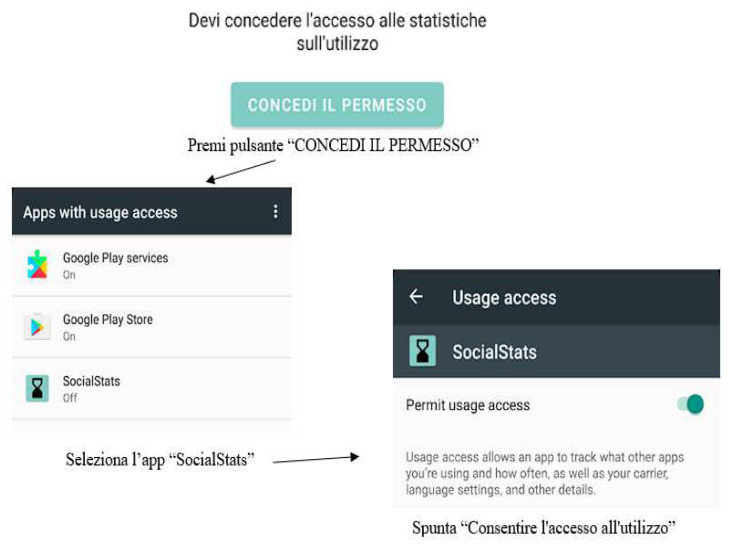
Permission phase of the app.

**Figure 2 ijerph-19-07445-f002:**
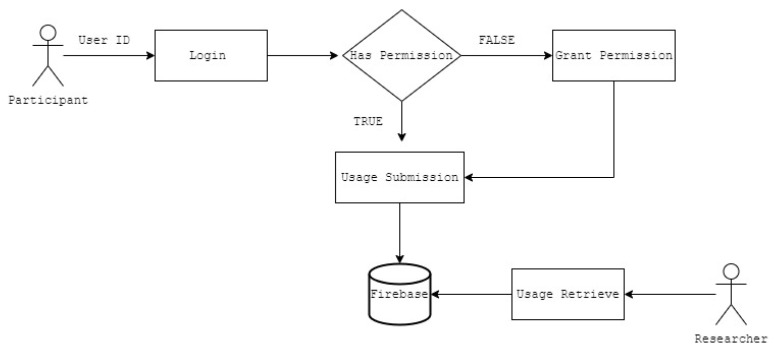
SocialStasApp Participant and Researcher workflow.

**Figure 3 ijerph-19-07445-f003:**
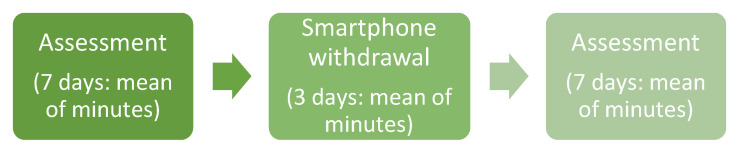
SocialStatsApp Research Design Schema.

**Figure 4 ijerph-19-07445-f004:**
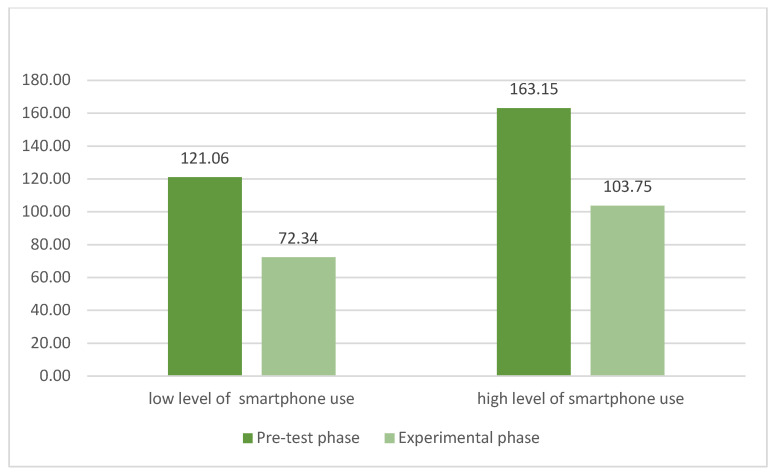
Percentage of participants with and without compliance in the two groups with high and low levels of smartphone addiction.

**Figure 5 ijerph-19-07445-f005:**
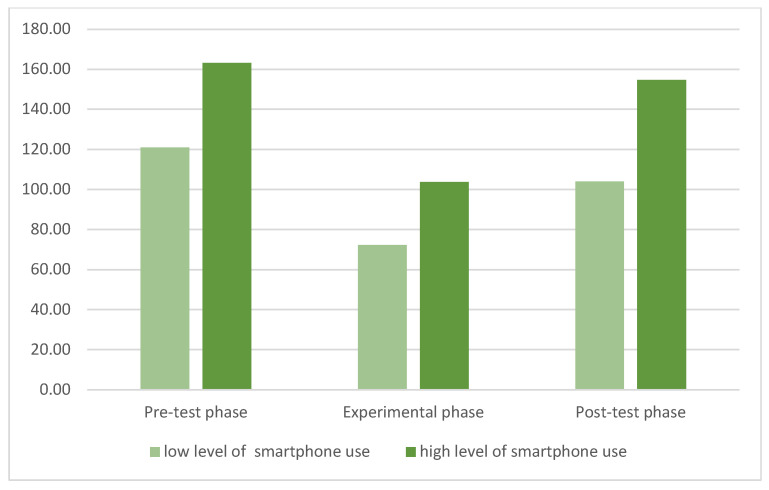
Means of total minutes of smartphone use in the three phases.

**Table 1 ijerph-19-07445-t001:** Means (and standard deviations) of total minutes of smartphone use in the three phases.

Groups	Pre-Test Phase	Experimental Phase	Post-Test Phase
Low levels of smartphone usage	121.06 (86.85)	72.34 (45.74)	103.98 (56.56)
High levels of smartphone usage	163.15 (67.60)	103.75 (57.97)	154.73 (55.60)

**Table 2 ijerph-19-07445-t002:** Means (and standard deviations) on cognitive tests of groups with low levels of smartphone usage and with high levels of smartphone usage (SU).

Groups	Pre-Test Phase	Post-Test Phase
**RT in Working Memory Test N-Back**		
Low levels of SU	1008.98 (212.08)	908.98 (253.34)
High levels of SU	1075.61 (237.98)	916.18 (129.53)
**RT in Visual Reaction Time**		
Low levels of SU	453.84 (132.09)	426.55 (111.56)
High levels of SU	429.76 (77.49)	430.39 (91.59)
**RT in Visual Reaction Time Go-No go**		
Low levels of SU	500.93 (154.64)	497.54 (214.90)
High levels of SU	534.44 (81.54)	533.46 (176.22)
**RT in Auditory reaction time**		
Low levels of SU	374.13 (117.40)	387.06 (112.06)
High levels of SU	387.15 (293.57)	374.48 (94.83)
**RT in Eriksen Flanker test**		
Low levels of SU	600.01 (143.14)	577.09 (140.73)
High levels of SU	646.02 (219.53)	558.10 (220.04)

**Table 3 ijerph-19-07445-t003:** Means (and standard deviations) on PGWBI, FoMOs and PS of the 2 groups: low levels of smartphone usage and high levels of smartphone usage.

Scale	High Levels of Smartphone Usage	Low Levels of Smartphone Usage
Psychological General Well-Being Index	5.38 (1.60)	6.20 (1.57)
Fear of Missing Out scale	2.36 (0.65)	1.87 (0.54)
Procrastination Scale	3.29 (1.36)	2.50 (1.13)

## Data Availability

Dataset is available from the link: https://data.mendeley.com/datasets/3s33tfd87m/1, accessed on 2 November 2021.

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
