# Peer review of "Problematic Smartphone Use Leads to Behavioral and Cognitive Self-Control Deficits"

_ijerph, 2022, doi:10.3390/ijerph19127445_

Round 1

Reviewer 1 Report

The  presented article can be positively valued, and accepted for publication.

Academic editor

In my opinion, the research topic is relevant, and the study is interesting. Parallelly, there are some issues that need to be addressed before the paper will be suitable for publication.

- The authors use the median in their sample to establish who has high or low levels of smartphone addiction. Why were the cut-offs considered in the Italian validation of the scale not used (De Pasquale, Sciacca, Hichy, 2017)? Perhaps it would be more appropriate.
- Throughout the text, there is a tendency to confuse smartphone addiction and social media use (and social media addiction). For example, the SocialStatsApp allows you to observe the time spent on some (not all) social media, and not the time spent using the phone. Yet in the text it is suggested that this is a measure that allows observing the time spent on smartphones (for example, this perception is very strong on page 3, lines 122-136). It would be important to review the text throughout the paper distinguishing these aspects well.

Author Response

- The authors use the median in their sample to establish who has high or low levels of smartphone addiction. Why were the cut-offs considered in the Italian validation of the scale not used (De Pasquale, Sciacca, Hichy, 2017)? Perhaps it would be more appropriate.

Thank you for your observation. The first reason is that the authors themselves underlined that their study has a limitation, the fact that the cut-off scores used for the Italian version of SAS-SV were based on the Korean version of the scale and the respective ROC analysis conducted in a sample of teenagers from Korea; since it was underlined as the only limitation of the work, we thought that the cut-off was not working.

The second reason is related to the fact that the aim of the present study was to divide the sample between high and low users of the smartphone and not select people with smartphone addiction.

- Throughout the text, there is a tendency to confuse smartphone addiction and social media use (and social media addiction). For example, the SocialStatsApp allows you to observe the time spent on some (not all) social media, and not the time spent using the phone. Yet in the text it is suggested that this is a measure that allows observing the time spent on smartphones (for example, this perception is very strong on page 3, lines 122-136). It would be important to review the text throughout the paper distinguishing these aspects well.

Thank you. We revised the entire text according to your suggestion.

This manuscript is a resubmission of an earlier submission. The following is a list of the peer review reports and author responses from that submission.

Round 1

Reviewer 1 Report

The main focus of this paper is to present the relationship between Smartphone addiction and therelationship with behavioral and cognitive self control deficits. The paper is written fairly. The current MS lacks in many aspects:

(a) justification is missing

(b) did not compare with stat of the art models

(c) lacking concrete analysis and discussion of results and models

(d) poor contribution. It seems that authors are linguistically as well computationally (Machine learning and deep learning )naive as the author skipped characteristics and challenges of mobile phones and skipped technical details of the models.
it is clear that the authors put some effort into this paper, but it is not clear that the effort was well directed.

Overall. This research requires a lot more rigor.

Below are general and detailed comments about the major issues
1. Lacks significant contributions from the authors
2. The authors adopted built-in models and datasets and performed simple experiments.

Author Response

FIRST REFEREE

The main focus of this paper is to present the relationship between Smartphone addiction and the relationship with behavioral and cognitive self-control deficits. The paper is written fairly. The current MS lacks in many aspects:

(a) justification is missing

*Reply

Thank you. In the first part of the “the present study” section we account for justification of the work.

(b) did not compare with state-of-the-art models

*Reply

Thank you. In the introduction section we added a full paragraph in which we explained the current status on the theories on smartphone and internet addiction.

(c) lacking concrete analysis and discussion of results and models

*Reply

Thank you. In the discussion section we added a analysis of results and models.

(d) poor contribution. It seems that authors are linguistically as well computationally (Machine learning and deep learning) naive as the author skipped characteristics and challenges of mobile phones and skipped technical details of the models. it is clear that the authors put some effort into this paper, but it is not clear that the effort was well directed.

*Reply

From a computational point of view, we entirely re-write the SocialStatsApp and the Technology Analysis.

From a linguistic point of view, we rewrite the work with the help of a mother-language specialist.

Reviewer 2 Report

the initially presented article can be positively valued, and accepted for publication, once a few small modifications have been made.
The font of the two figures should be revised, not formatted.
Within the section called "Method" a subsection that faithfully collects the research objectives as well as the hypotheses proposed would be appreciated.

Author Response

SECOND REFEREE

The initially presented article can be positively valued, and accepted for publication, once a few small modifications have been made.
The font of the two figures should be revised, not formatted.

*Reply

Thank you. We did it

Within the section called "Method" a subsection that faithfully collects the research objectives as well as the hypotheses proposed would be appreciated

*Reply

Thank you. In relation to your suggestion, we added a full paragraph on the link between hypotheses and research objectives:

In relation to the main hypothesis that people with high levels of smartphone addiction may fail to reduce the time spent on the smartphone, we divided participants by median split into two groups to identify those with low levels (below median) and high levels (above median) of smartphone use and calculate their mean of minutes of use of it in three phases: pre-test phase, an experimental phase (with smartphone limitation) and a post-test phase (Figure 2).

In relation to the second hypothesis that people with high levels of smartphone use may show deficiencies in cognitive tests, we submitted the two groups to working memory n-back test, visual reaction time, auditory reaction time, visual reaction time, go/no-go test and Eriksen flanker test.

In relation to the third hypothesis that people with high levels of smartphone use may show a lower Psychological General Well-Being Index, a higher Fear of Missing Out score and Procrastination score than participants with low level of smartphone use, we asked to both groups to fill the related questionnaires.

Reviewer 3 Report

Comments and Suggestions for Authors

Thank you for the opportunity of reviewing this interesting article. The main objective of the present study was to investigate whether participants with  325 high levels of smartphone addiction may have difficulty in self-control compared to participants with low levels of smartphone addiction.  This approach seems to be extremely relevant and promising.

Comments and Suggestions for Authors:

  • Please describe in detail how your study fits for aims and scope of International Journal of Environmental Research and Public Health (IJERPH).
  • As the authors themselves point out a limitation of the present study is related to the sample. Based on such limited material, it is difficult to draw reasonable conclusions that will cause complete confidence.
  • Please explain why the short version of the Psychological General Well-Being Index (PGWBI) was used in this study but not  the WHO-5 Wellness Index developed by the Psychiatric Research Unit, WHO Collaborating Center for Mental Health, Frederiksborg General Hospital, DK-3400 Hillerød.Please describe in more detail the tools you used in the research.
  • Conclusions section should be presented as separate part.
  • Please improve the structure of the article in accordance with the Instructions for Authors.
  • For theoretical framework and bibliography additional current references should be included to new research

Author Response

THIRD REFEREE

Thank you for the opportunity of reviewing this interesting article. The main objective of the present study was to investigate whether participants with high levels of smartphone addiction may have difficulty in self-control compared to participants with low levels of smartphone addiction.  This approach seems to be extremely relevant and promising.

Comments and Suggestions for Authors:

  • Please describe in detail how your study fits for aims and scope of International Journal of Environmental Research and Public Health (IJERPH).

*Reply

Thank you. We think that in a changing world in which every daily activity is related to the use of smartphones, it’s necessary to try understanding the effects of these devices on individuals. From beginnings until now, smartphones have faced an important evolution. That made more simple many activities, in different areas, such as the social and work ones. For these reasons, we believe that it’s important, in a society where smartphone is so used, to analyze which consequences smartphones could have on public health.

  • As the authors themselves point out a limitation of the present study is related to the sample. Based on such limited material, it is difficult to draw reasonable conclusions that will cause complete confidence.

*Reply

You are right. We think that the most addicted participants simply refuse to adhere to the research. We described this both in the participant and in the limitation sections. Since we obtained anyway that participants with high level of smartphone use were not able to refrain from its usage, we think it may be a useful indication of withdrawal symptoms. We better describe this in the related section.

  • Please explain why the short version of the Psychological General Well-Being Index (PGWBI) was used in this study but not the WHO-5 Wellness Index developed by the Psychiatric Research Unit, WHO Collaborating Center for Mental Health, Frederiksborg General Hospital, DK-3400 Hillerød. Please describe in more detail the tools you used in the research.

*Reply

The Psychological General Well-Being Index (PGWBI) turns out to be an excellent self-reported measure of health-related quality of life and health-related quality of life. Cronbach’s value of PGWBI scale is between 0.90 and 0.94, meaning that this tool is valid. Moreover, there are many studies in which PGWBI were used to evaluate the impact of medical-surgery or psychological therapies on life qualify. World Health Organisation- Five Well-Being Index (WHO-5) is a unidimensional scale that estimates subjective psychological well-being using only 5 items. Considering that in this study more dimensions and aspects of individual well-being and life quality have been evaluated, it would have been reductive to use this tool. On the contrary, the PGWBI, with its 22 items, allows the evaluation of 6 dimensions, making it more suitable for the present research.

  • Conclusions section should be presented as separate part.

*Reply

Thank you, we added the full paragraph as a separate part.

  • Please improve the structure of the article in accordance with the Instructions for Authors.

*Reply

Thank you. We did it.

  • For theoretical framework and bibliography additional current references should be included to new research

*Reply

Thank you. We added new current references.